# Experimental Methodology to Characterize the Noise Paths in a Horizontal-Axis Washing Machine

Cristian Albero * and Beatriz Sánchez-Tabuenca

Mechanical Engineering Department, University of Zaragoza, María de Luna 3, 50018 Zaragoza, Spain; bstb@unizar.es
* Correspondence: calbero@unizar.es

**Abstract:** In this paper, an experimental methodology to characterize the noise paths in a washing machine with a horizontal axis was developed. The noise paths considered in this research were the noise that escapes through holes, the non-resonant path through the panels, and the noise radiated by the panels of the cabinet. The characterization method was based on several sound intensity measurements on the outside panels of the washing machine. In addition to these measurements, characterization of the radiation factor was performed by applying a method that relates intensity and vibration measurements while the structure of the washing machine is excited using a shaker. Applying the methodology to a washing machine, the main transmission path of the noise along the frequency domain where this home appliance has its highest values was identified. This methodology can provide the manufacturer with a guide to improve the acoustic performance of washing machines by applying noise control solutions in the noise path depending on the frequency domain.

**Keywords:** washing machine; noise paths; intensity measurements; vibration velocity measurements

## 1. Introduction

Reducing noise levels in household appliances, particularly washing machines, has become a top priority for manufacturers due to increased user demands for home comfort. Noise has become a decisive factor in purchasing decisions since washing machines are one of the household appliances that emit the highest levels of noise [1]. To address this issue, there are established standards for the acoustic and vibration performance of washing machines [2]. Furthermore, noise exposure can pose a serious threat to the health of users [3].

Several authors have focused their research on the study of the noise of household appliances by characterizing sound quality indexes [4,5]. However, noise reduction methods can be expensive and challenging to apply, forcing manufacturers to compromise on durability, efficiency, and cost.

There are several techniques available for noise control, but this does not necessarily imply the reduction of noise emissions from the sources. Instead, it is possible to apply techniques to reduce noise transmission and propagation, and protection techniques for the receivers [6]. In the field of home appliances, only the first two types of techniques are applicable.

Research on the use of green materials applied to the noise control of home appliances was conducted by Fatima and Mohanty [7].

The characterization of sources in washing machines using different methods has been investigated by several authors, including Barpanda and Tudor, who focused their research on changes in the sound pressure level under different operating conditions [8]. Other authors applied the transfer path analysis technique to identify sources and their contributions to exterior noise [9,10].

Chiarotti et al. conducted a study on the acoustic emission of a washing machine using a 30-microphone array to identify areas with the highest acoustic emission from

the appliance [11]. Moreover, Koizumi et al. analyzed the noise radiated by the panels of a washing machine based on their vibration levels using simulations and experimental characterizations [12].

The objective of this research was to develop a methodology that characterizes and quantifies the noise paths in a horizontal-axis washing machine. The noise generated by a washing machine results from various elements and is transmitted through different noise paths from the appliance's interior to the exterior of its cabinet, which is perceived by the user. The washing machine's acoustic performance must be optimal under different operational conditions, such as varying spinning speeds and laundry loads; thus, the distributed load and the position and magnitude of the unbalance created must be considered. Hence, it is essential to have a methodology that can efficiently and accurately characterize the noise paths under multiple conditions without significant effort related to measuring.

The proposed method is based on a few experimental intensity and vibration measurements, providing a guide for acousticians to assess and improve the washing machine's acoustic performance. This methodology can help manufacturers to design washing machines with better acoustic performance and meet the users' expectations.

This paper is organized as follows: In Section 2, the washing machine under study is presented and the different noise sources and transmission paths are described. In addition, the experimental measurement methods are described in this section. Section 3 describes the power values obtained, and the characterization of each of the noise paths. Finally, the conclusions of the research are presented in Section 4.

This research introduces innovative aspects related to the characterization of noise emitted by a horizontal washing machine, specifically, on the transmission paths of the noise, including the noise that escapes through holes, the non-resonant path, and the noise emitted by the cabinet through radiation. Understanding these transmission mechanisms is crucial for acousticians to apply appropriate control treatments. The characterization of different noise transmission paths has not been studied by any previous authors. While some studies have identified noise sources and their contribution to total noise, such as Karsen [9] and Wang [10], other authors analyzed the noise emitted by home appliances under various conditions [8], and some present methods for reducing noise through different control treatments, such as Fatima in his research [7]. However, none of these studies delve into the power contributions of each transmission path.

The impact of the holes of the cabinet on the noise generated by the washing machine, particularly in the mid and high frequency ranges, is not addressed in the existing literature. Effective reduction of noise in these frequencies requires implementation of noise control treatments on gaps and openings, while structural modifications to the cabinet are unlikely to significantly improve the acoustic performance of the appliance.

This paper presents a precise method for determining the radiation factor and noise emitted by washing machine panels. Although similar methods have been applied to other systems as Kozupa describes in his work [13], the application of this technique to a washing machine is not addressed in the existing literature. In addition to characterizing the radiation contribution to the overall power, this method allows acousticians to identify the zones of panels with the highest radiated emission for various appliance configurations using only vibration measurements. This approach is simpler and faster than noise measurements, providing a more efficient means of designing home appliances.

The study concludes by presenting the distinct contributions of each transmission path on either side of the washing machine. This significant insight could prove highly valuable for acousticians, enabling them to implement targeted interventions on specific transmission paths and improve overall acoustic performance.

## 2. Materials and Methods

### 2.1. Washing Machine

The appliance selected for the research was a front-load washing machine with a horizontal axis and a laundry capacity of 9 kg. It comprised a plastic tub suspended from the housing by three springs and three free-stroke friction shock absorbers at the bottom. To stabilize its movement during the spinning phase, the machine had two front counterweights and an additional counterweight at the top. A ring-shaped gasket attached the tub along its perimeter to the front panel of the housing to prevent water leakage and two small gaskets for the inlet and the outlet of the water. Inside the tub, there was a drum that was supported by two bearings at the back of the shaft. The drum's motion was generated by a brushless direct current motor that connects to the drum shaft through a two-pulley system.

The housing of the appliance comprised four vertical steel panels and a steel panel at the bottom, connected with screws and TOX points. On the front side, there was one main hole with the door of the washing machine, and a plastic small door to allow access to the pump. The top of the front panel contained the detergent dispenser and electronic controls.The top panel of the cabinet was made of wood with a thickness of 15 mm. The cabinet's top panel was made of 15 mm thick wood. The back and bottom panels had multiple holes for fixing various elements during configuration and transportation. It is worth noting that there were gaps between the panels, which were not sealed for acoustic transmission.

The present investigation focused on analyzing the effects of an unbalanced load of 600 g, generated using a lead plate, with the drum rotating at a constant speed of 1400 rpm.

Figure 1 illustrates the schematic representation of the elements that generate noise and the potential pathways through which the noise can be transmitted.

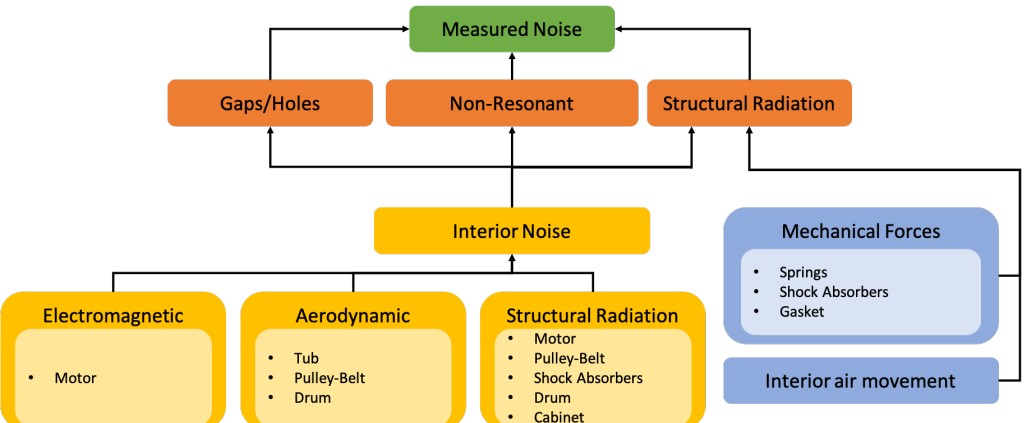

**Figure 1.** Noise sources and acoustic paths in a washing machine.

In order to effectively analyze the acoustic performance of a washing machine, it is crucial to first gain a thorough understanding of the various types of noise generated within its cabinet. This study focuses on three main sources of noise: structural, aerodynamic, and electromagnetic.

The first source of noise arises from the structural radiation of components within the washing machine, including the electric motor, transmission components, such as the pulleys and belt, shock absorbers, drum, and bearings, and the cabinet vibration [14,15]. The acoustic emission produced by these elements can significantly contribute to the overall noise level of the washing machine.

Aerodynamic noise is the second source of noise studied in this research, which is generated by the movement of the oscillating group due to unbalanced loads within the drum. This movement can cause air turbulence, leading to a variety of noise types [16].

The final source of noise is electromagnetic in nature and results from the structure of the motor and its electromagnetic excitation [17]. This type of noise can be a significant contributor to the overall noise generated by the washing machine.

Taken together, these three sources of noise comprise the interior noise of the washing machine, which can be transmitted to the exterior through various noise paths or dissipated within the appliance.

The extent to which interior noise is transmitted to the exterior is primarily determined by the structural properties of the cabinet. Long conducted an extensive study on the transmission of noise across various frequencies in the presence of diffuse incident noise using different approaches to characterize structures [18].

The first transmission path considered between the interior and exterior of a washing machine is through the holes and gaps in the cabinet, and is influenced by the magnitude of interior noise and the shape and surface of the holes.

Furthermore, noise can also be transmitted through the panels of the structure in different ways depending on its frequency, which is characterized by the transmission loss of the surrounding structure.

To comprehensively assess the acoustic characteristics of a washing machine, it is imperative to evaluate its transmission loss type, which requires one to determine the frequency of the first mode and the critical frequency of the panels in order to define the limits of the mass law region. The first mode of the panels of the cabinet of the washing machine analyzed was observed at 30 Hz. Furthermore, estimating the critical frequency of the wavelength can be accomplished by utilizing the following equation, where $c$ is the speed of sound in air, $m$ is the surface density, $\sigma$ is the Poisson ratio, $E$ is the Young's modulus, and $h$ is the panel thickness. The values of these parameters for each of the panels are shown in Table 1.

$$f_c = \frac{c^2}{2\pi} \sqrt{\frac{12m(1-\sigma)}{Eh^3}} \tag{1}$$

**Table 1.** Structural parameters of the panels of the cabinet and their critical frequencies.

| Plate | $c$ [m/s] | $m$ [kg/m$^2$] | $\sigma$ | $E$ [N/m$^2$] | $h$ [m] | $f_c$ [Hz] |
|---|---|---|---|---|---|---|
| Front | 340 | 7.02 | 0.3 | $2.0 \times 10^{11}$ | 0.0009 | 12,047 |
| Right/Left | 340 | 6.24 | 0.3 | $2.0 \times 10^{11}$ | 0.0008 | 13,553 |
| Back | 340 | 7.80 | 0.3 | $2.0 \times 10^{11}$ | 0.001 | 10,843 |

The frequency domain examined in this study ranged from 125 Hz to 8000 Hz, encompassing the primary noise range of the washing machine. Based on these values, it can be inferred that the dominant mode of transmission for the interior noise through the panels was via the non-resonant path.

The last type of noise that is emitted externally from a washing machine is radiated noise. This is caused by the motion of the cabinet generated by the various elements of the washing machine that are fixed to the panels, including springs, shock absorbers, and gaskets that seal the tub. While the acoustic coupling in the frequency domain may not be particularly strong, as previously discussed, the vibration level of these panels can be high, depending on the laundry load and its unbalance magnitude in the washing machine. Therefore, it is important to consider the impact of this radiated noise when assessing the overall noise level of the washing machine.

Figure 2 presents an outline of the different transfer paths analyzed.

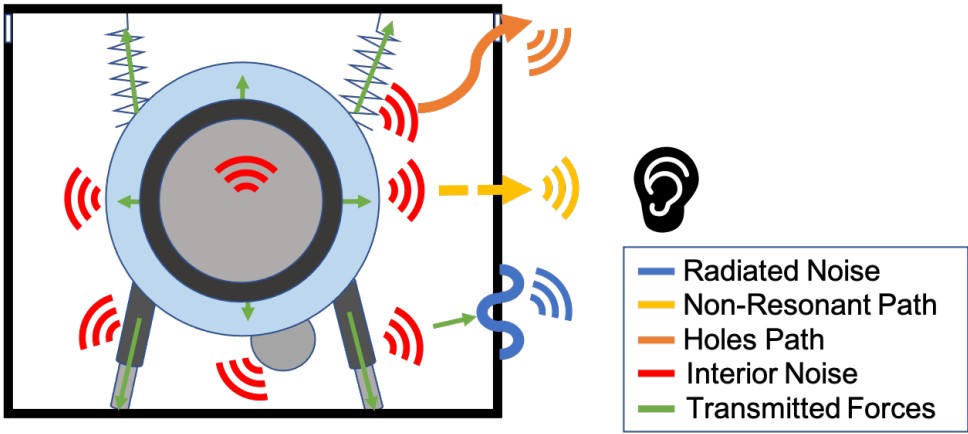

**Figure 2.** Scheme of noise transmission paths.

*2.2. Power Measurements*

The acoustic measurements were conducted in an acoustic chamber with absorbent walls to minimize external noise interference. The washing machine was positioned at the center of the room to ensure that the measurements captured the acoustic characteristics of the machine accurately. The background noise level in the chamber was at least 20 dB below the operating noise level of the washing machine, in compliance with established standards.

The data acquisition system used in this study consisted of a multi-analyzer Pulse 3160-A-042, which was controlled by a computer running the PULSE LabShop 9.0 software from the company Hottinger Brüel & Kjær. Sound intensity measurements were obtained using a 3599 Brüel & Kjær sound intensity probe . A schematic representation of the experimental setup is provided in Figure 3a.

The intensity measurements were conducted in accordance with ISO standard ISO/TS 7849-1:2009 [19]. Each side of the washing machine was measured separately, including the front, right, back, left, and top, with a distance of 15 cm maintained around each side. The measurement process lasted for 2 min per side and was carried out in the sequence illustrated in Figure 3b. Octave bands ranging from 125 Hz to 8000 Hz were used to register the measurements.

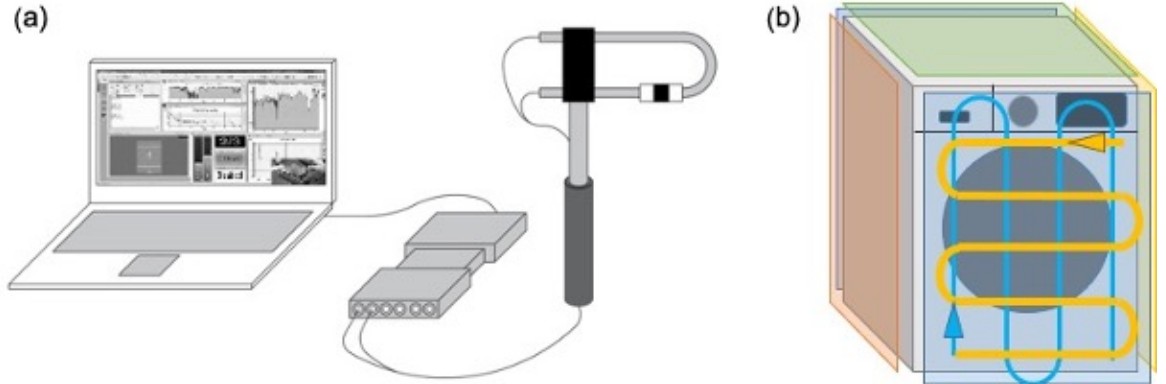

**Figure 3.** Intensity measurements: (**a**) measurement equipment; (**b**) measurement sequence.

With the intensity values, it was possible to obtain the power emitted by each of the sides of the washing machine using the following equation, where $\bar{I}$ is the average intensity,

$S$ is the surface of the side, $P$ is the power measured, and $i$ is the name of the side. Those power values were added to obtain $P_{Measured}$.

$$P_i = \int I dS = \bar{I}_i S_i \tag{2}$$

$$P_{Measured} = P_{Front} + P_{Right} + P_{Back} + P_{Left} + P_{Top} \tag{3}$$

The power was converted to dB by applying the following equation and the A-weight, obtaining $L_{w_{Measured}}$, where $P_0$ is the reference power, 1 pW [19].

$$L_{w_{Measured}} = 10 \log \left( \frac{P_{Measured}}{P_0} \right) \tag{4}$$

*2.3. Radiation Factor*

The present study aimed to characterize the radiated power emitted by the washing machine's structure by relating velocity and intensity measurements. This methodology has been previously employed by various researchers for characterizing non-homogeneous structures, as evidenced in the works of Kozupa et al. [13]. This method is also defined in accordance with ISO/TS 7849 [20], which lends it a high degree of reliability and validity. The application of this approach to the present research problem was expected to yield accurate and informative results regarding the radiated power emanating from the washing machine's structure. The power radiated by a vibration structure is defined by the following equation, where $P_i$ is the radiated power, $\rho$ is the air density, $c$ is the propagation speed of the sound in air, $v_i^2$ is the square velocity of the area, $S_i$ is the area, and $\sigma_i$ is the radiation factor of the surface analyzed.

$$P_i = \rho c v_i^2 S_i \sigma_i \tag{5}$$

The same equation in dB can be expressed as follows, where $S_0$ is the surface reference 1 m$^2$ and $(\rho c)_0$ is the reference 400 Ns/m$^3$.

$$L_{w_s} = L_{v_s} + 10 \log \frac{S_s}{S_0} + 10 \log \sigma_s + 10 \log \frac{\rho c}{(\rho c)_0} \tag{6}$$

The radiation factor was obtained by analyzing the relation between the intensity and the velocity of certain areas under a controlled excitation of the washing machine. The excitation was applied using an electromagnetic shaker coupled to the structure of the washing machine. The excitation signal was defined as a flat random spectrum from 100 Hz to 8000 Hz. The acquisition equipment consisted of an accelerometer 4397 Brüel & Kjær, an intensity probe 3599 Brüel & Kjær, and an electromagnetic shaker TiraVib, as is shown in Figure 4a. Motion measurements captured with the accelerometer were taken in front of the intensity probe, while the shaker was connected to the washing machine structure using a circular 0.5 m beam to disassemble the eigenmodes of the shaker, as illustrated in Figure 4b. The measurements were recorded in one-third octave bands ranging from 100 Hz to 8000 Hz to establish a more accurate relationship between the excited modes and their radiation

The panels of the washing machine were manufactured by stamping and bending processes and their shape was non-uniform. In order to accurately determine the radiation factor of each panel zone, an experimental setup was devised involving the placement of an accelerometer at the center of each zone and an intensity probe located 15 cm in front of it. The panels were divided into 86 different sections, as depicted in Figure 5.

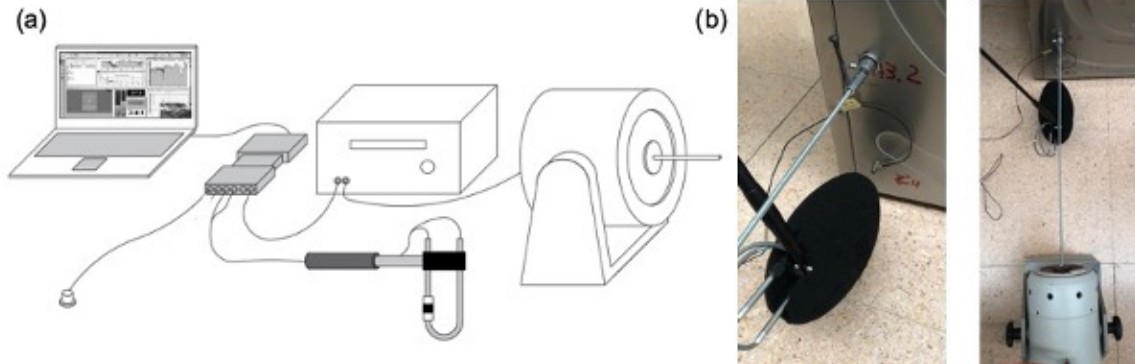

**Figure 4.** Radiation factor measurements: (**a**) measurement equipment; (**b**) measurement setup.

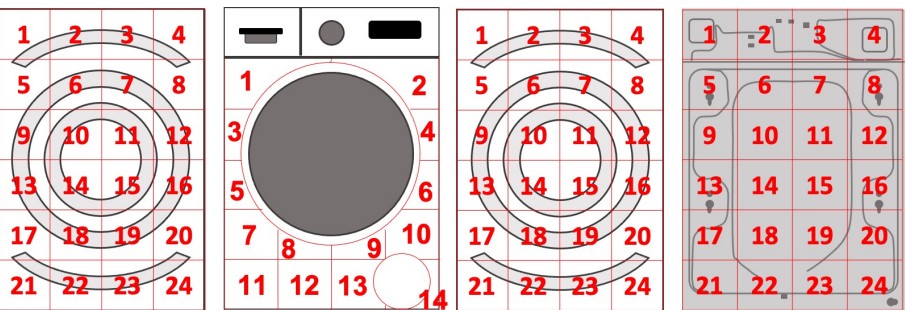

**Figure 5.** Division of the panels.

To determine the radiation factor of a structure, the following equation can be used, which requires the previously described measurement of the intensity and velocity of each zone.

$$\sigma_s = \frac{I_s}{\rho c v_s{}^2} \tag{7}$$

The shaker excitation was applied at six distinct points on the panel of the washing machine, and the resulting measurements were recorded for each point. The radiation factor was calculated by averaging the six measured values for each point on the panel.

### 2.4. Vibration Measurements

To obtain the radiated power of each defined zone, it is imperative to measure the vibration velocity level $L_{v_s}$ of each zone under operational conditions. The vibration can be obtained using the following equation, where the reference velocity $v_0$ is 50 nm/s:

$$L_{v_s} = 10 \log \left( \frac{v_s^2}{v_0^2} \right) \tag{8}$$

The equipment utilized for the measurements comprised a pair of 4397 Brüel & Kjær accelerometers connected to the previously described data acquisition system. The measurements were obtained in third-octave bands, i.e., using the same methodology employed for obtaining the radiation factor.

### 3. Results

#### 3.1. Experimental Measurements

The sound intensity probe was utilized to measure the power of each side of the washing machine in the standard configuration. The power values acquired are presented in Figure 6.

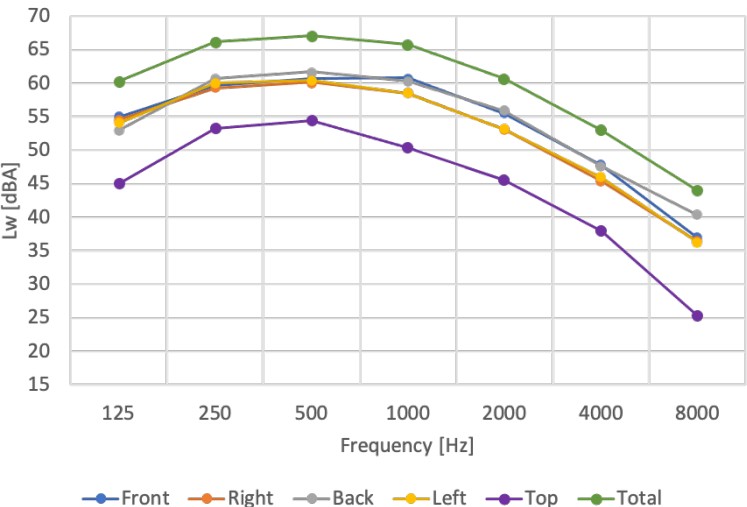

**Figure 6.** Measured power of standard washing machine.

The power emitted by the washing machine can be split in the following contributions, where $P_{Standard}$ is the total power measured, $P_{Holes}$ is the power transmitted to the exterior through the holes and leaks in the cabinet, $P_{Radiated}$ is the power radiated by the panels of the washing machine, and $P_{Non-Resonant}$ is the contribution that is transmitted through the non-resonant path.

$$P_{Standard} = P_{Holes} + P_{Radiated} + P_{Non-Resonant} \tag{9}$$

To measure the power transmitted through the holes of the structure of the washing machine accurately, we employed a compressed poro-elastic material to cover the gaps between the panels of the cabinet and the holes in the structure. The washing machine was tested with the holes covered, and the resulting data are presented in Figure 7.

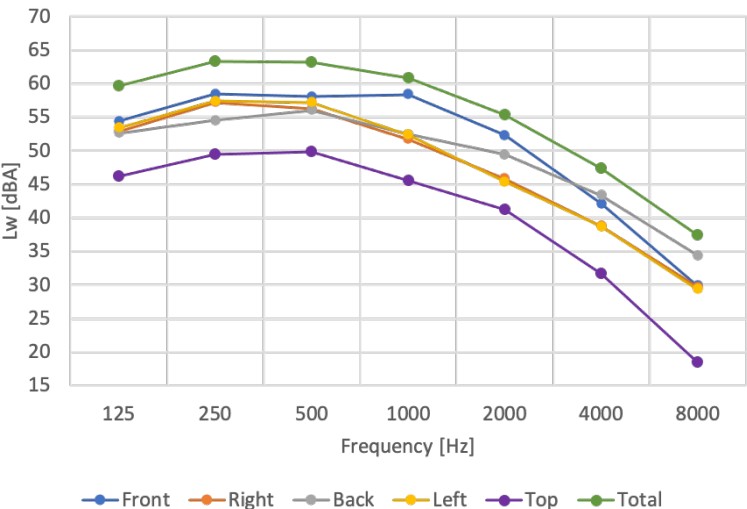

**Figure 7.** Measured power of washing machine without holes.

The power measured in this appliance with holes covered is named $P_{Covered}$. It is assumed that the incident noise inside was constant in both configurations, and the suppression of the transfer path of the noise escaping through the holes did not modify any of the other paths. The different noise paths of this configuration are defined by the following equation:

$$P_{Covered} = P_{Radiated} + P_{Non-Resonant} \tag{10}$$

From the previous equations, $P_{Holes}$ can be defined by the subtraction of the power of the two configurations according to the following equation:

$$P_{Holes} = P_{Standard} - P_{Covered} \qquad (11)$$

The overall levels of these two power measurements and $P_{Holes}$ converted to dBA are shown in Table 2 in dBA for each side of the appliance.

**Table 2.** Power of standard washing machine.

| Face | $Lw_{Standard}$ [dBA] | $Lw_{Covered}$ [dBA] | $Lw_{Holes}$ [dBA] |
|---|---|---|---|
| Front | 66.1 | 64.0 | 61.9 |
| Right | 64.9 | 61.2 | 62.5 |
| Back | 66.4 | 60.6 | 65.1 |
| Left | 65.2 | 61.8 | 62.6 |
| Top | 58.2 | 54.4 | 55.9 |
| Total | 72.0 | 68.3 | 69.4 |

The radiated path was defined by characterizing the radiation factor of the washing machine using the methodology outlined in this paper. The radiation factor values were determined based on the velocity and radiated intensity measurements obtained from the steel panels of a standard washing machine under controlled excitation. Figure 8 illustrates the radiation factor values for the right panel, as an example, across each third octave.

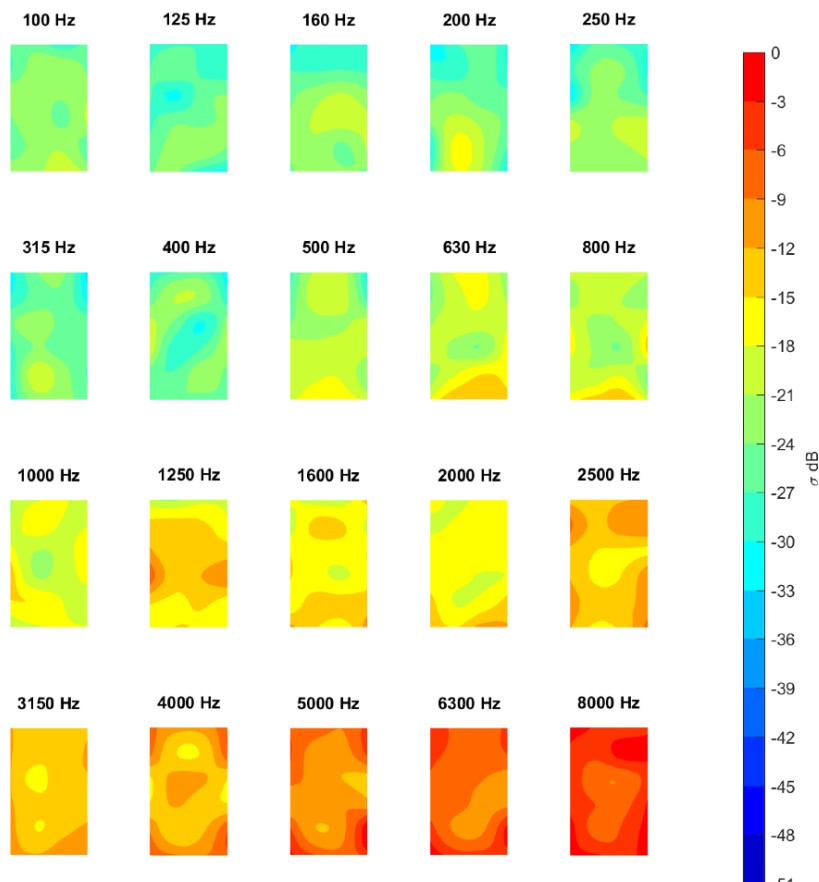

**Figure 8.** Radiation factor of the right panel.

In addition, the velocity of each zone of the panels was measured while the washing machine was spinning at 1400 rpm. The obtained values of $L_v$ for the right panel are plotted

in dBA with a reference of 50 nm/s in Figure 9a. The intensity radiated by each of these zones can be determined by applying the following equation, which is illustrated for the right panel in Figure 9b.

$$L_{I_s} = L_{v_s} + 10\log\sigma_s + 10\log\frac{\rho c}{(\rho c)_0} \tag{12}$$

To calculate the power of each zone $L_{w_s}$, the following equations can be used. The total radiated power of each side of the structure can then be obtained by taking the logarithmic sum of these values. The radiated power is presented in Figure 10 in octaves, while Table 3 shows the total values in dBA for each side.

$$L_{w_s} = L_{I_s} + 10\log\frac{S_s}{S_0} \tag{13}$$

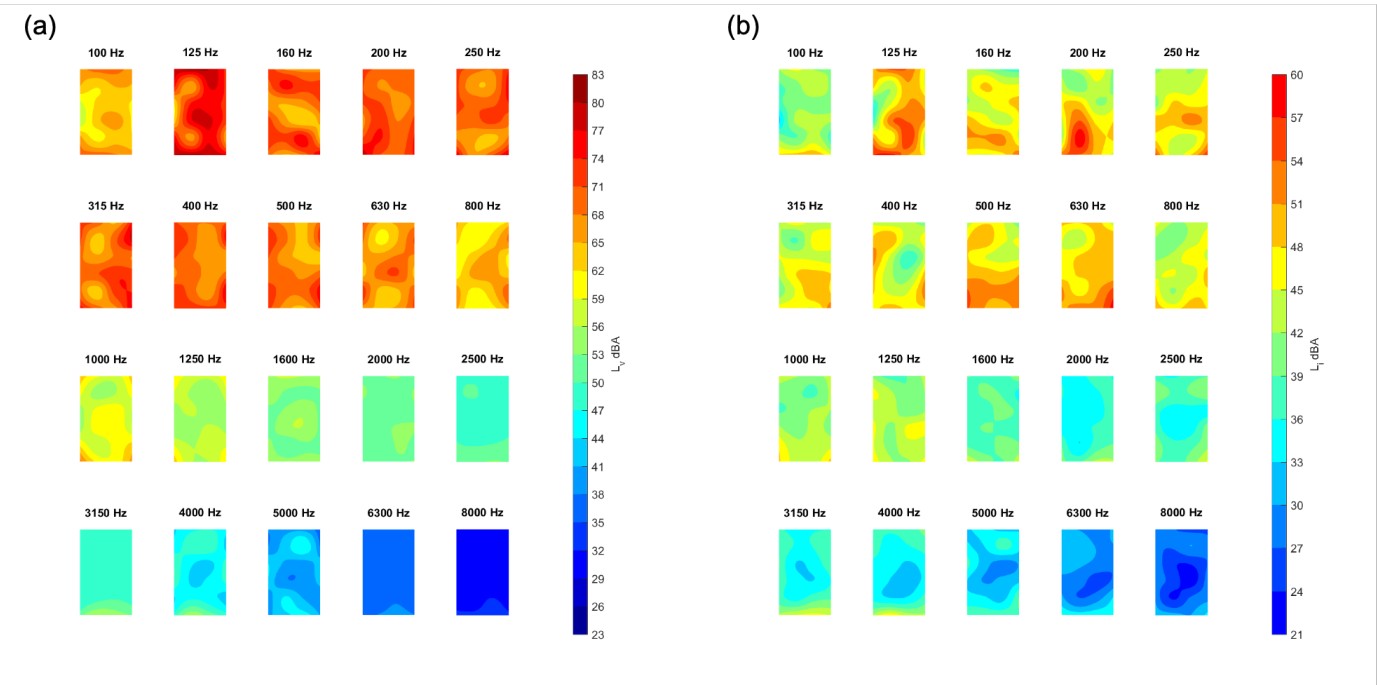

**Figure 9.** (**a**) Velocity measurements of right panel; (**b**) intensity radiated of right panel.

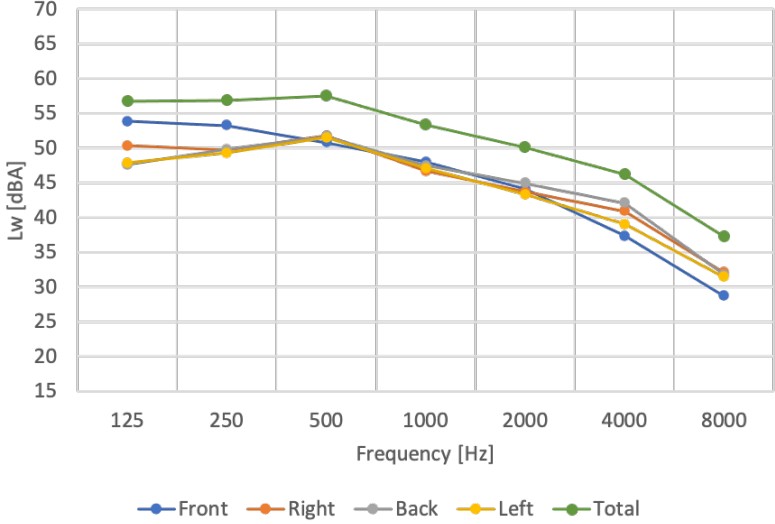

**Figure 10.** Radiated power.

**Table 3.** Radiated power.

| Face | $Lw_{Radiated}$ [dBA] |
|---|---|
| Front | 58.3 |
| Right | 56.4 |
| Back | 56.3 |
| Left | 55.7 |
| Total | 62.8 |

After characterizing the power levels, it became possible to determine the non-resonant path ($P_{Non-Resonant}$) of the equation by subtracting the other two transmission powers from the total power of the standard washing machine, according to the following equation:

$$P_{NonResonant} = P_{Standard} - P_{Radiated} - P_{Holes} \qquad (14)$$

*3.2. Noise Paths*

Figure 11a illustrates the power of each transmission path, while Figure 11b displays the percentage contribution of each octave to the total power.

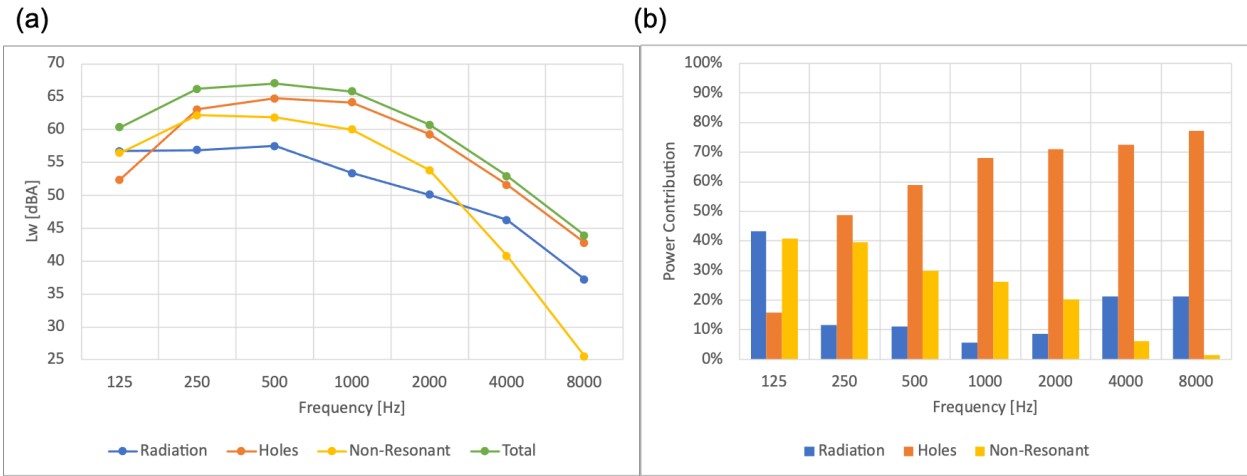

**Figure 11.** (**a**) Transmission path power levels; (**b**) transmission path contributions.

After analyzing the data, it is evident that the most significant factor contributing to noise transmission was the escape of sound through the structure's gaps and holes. It should be noted that this contribution was negligible in the first octave analyzed. However, as the frequency increased, the power transmitted through this noise path became increasingly significant, accounting for 16% of the total power in the first octave and escalating to a staggering 77% in the highest frequency range.

The non-resonant path had a greater impact on lower frequencies, but its influence decreased as the frequency increased. The maximum power transmitted through this path occurred at 250 Hz, with a power of 62.2 dBA. The most substantial contribution in terms of percentage was at 125 Hz, with 41% of the total power at that octave. In the washing machine under study, the radiation levels were lower than the other two noise paths. The primary contribution of this path was in the first octave, 125 Hz, with a power level of 56.7 dBA, representing 43% of the total noise from the washing machine. At 1000 Hz, the contribution from this path was minimal, accounting for only 6% of the overall power.

After examining the contributions obtained, it was possible to identify the appropriate noise path to address based on the frequency.

In order to achieve a reduction in power at lower frequencies, two potential solutions can be considered: minimizing transmission through the non-resonant path, which was the primary source of noise in the 125 Hz octave, or decreasing the power radiated by the panels of the washing machine. The reduction of noise through the non-resonant path can be

approximately quantified by the mass law, which is formulated with the following equation for a diffuse acoustic field with a maximum wave incidence of 78°, where $\Delta L_{TL_{MassLaw}}$ is the transmission loss of the mass law, $\omega$ is the frequency in radians per second, $m_s$ is the surface mass density of the panel material, $\rho_0 c_0$ is the acoustic impedance of the air, $f$ is the frequency in Hz, and $K_{TL}$ is a numerical constant with a value of 47.3 dB when the equation is used in metric units [18].

$$\Delta L_{TL_{MassLaw}} = 10 \log \left[ 1 + \left( \frac{\omega m_s}{3.6 \rho_0 c_0} \right)^2 \right] = 20 \log \left( f m_s \right) - K_{TL} \tag{15}$$

Based on the equation, it can be observed that every time the surface mass density of the panels is doubled, there is a reduction of 6 dB in the power transmitted.

To reduce the noise generated by the vibration of the panels in a washing machine, it is essential to take action on the forces transmitted by the connecting elements that link the cabinet to the oscillating group of the appliance. These connecting elements include the water inlet and outlet gaskets, the front gasket that seals the mouth of the drum with the door, and the dampers that dissipate energy during the spinning cycle of the drum when the rotation frequency coincides with the natural frequencies of the rigid group.

However, it is important to note that reducing these forces could result in an increase in the displacement of the washing machine during the acceleration phase. Therefore, it is necessary to find a compromise that effectively minimizes noise radiated while minimizing any negative effects on the washing machine's performance.

The data reveal that the washing machine generated the most significant amount of acoustic emission in the frequency range of 250 Hz to 2000 Hz. While the contribution of radiation in this range was relatively insignificant, the noise that escaped through the holes accounted for a substantial portion of the overall emission, compared to the non-resonant path. To mitigate the transmission of noise through the frame holes, covering them could be an effective solution, as the extent of transmission was closely related to their area and shape, as stated in Pereira's research [21].

The noise level emitted by a washing machine was not significant at high frequencies, particularly in the octaves of 4000 and 8000 Hz. The non-resonant path had a minimal influence, while radiation accounted for approximately 21% of the noise emitted. Since these frequencies were close to the critical frequency of the panels, the interior noise could significantly excite the panels, whereas the excitation of connected elements was lower due to the excitation spectrum of mechanical forces being higher at lower frequencies. The remaining 75% of noise was transmitted through the holes.

Additionally, Figure 12a illustrates the individual contributions of the three transmission paths for each side of the washing machine, and the total. Moreover, Figure 12b displays the contribution of each side to the overall power of the transmission paths.

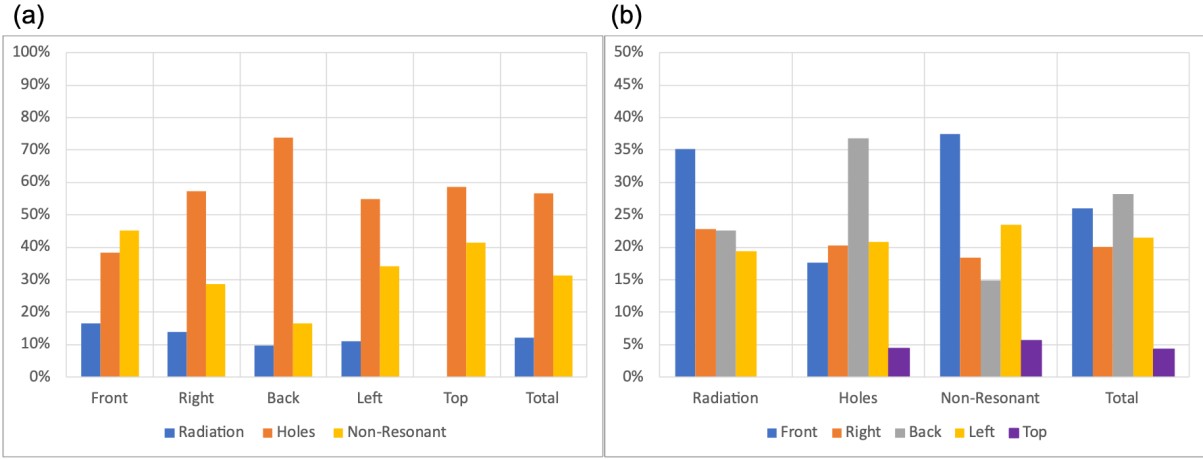

**Figure 12.** (**a**) Path contribution to each side; (**b**) side contribution to each path.

The results reveal similar noise behavior between the right and left panels, with noise through holes representing 57% and 55% of the total noise, respectively. Additionally, noise transmitted through the non-resonant path accounted for 29% and 34% of the total noise of those sides.

On the front side of the washing machine, the contribution of the non-resonant path was 45%, while radiated power and noise through holes represented 17% and 38%, respectively. The high contribution of the non-resonant path can be attributed to the lower surface density of the plastic parts that comprised this side, in contrast to the steel panel. Moreover, the higher contribution of the radiated noise was caused by the excitation of the front gasket that connected the tub and the cabinet, accounting for 35% of the radiated noise.

Analyzing the data for the back side, the holes had the highest contribution, accounting for 74% of the total noise through this side and 37% of the total noise. This behavior can be explained by the number and area of the holes present in this panel.

Regarding the top side, the noise contribution was relatively low compared to the other sides of the washing machine, accounting for only 4% of the total noise, with 41% due to the non-resonant path and 59% to holes.

Furthermore, an analysis of the contribution of each path in the total noise reveals a dominance of the holes transmission path, representing 57%. In contrast, the non-resonant path accounted for 31%, and radiated noise contributed 12%. Examining the contribution of each side of the washing machine, the front, right, back, and left sides contributed similarly, with 26%, 20%, 28%, and 21%, respectively. In contrast, the top panel's contribution to the total noise was only 4%.

## 4. Conclusions

The primary contribution of this study is the development of a novel methodology to characterize and quantify different noise paths emitted by a washing machine. This was achieved by measuring the experimental intensity and vibration data for two configurations: the standard appliance and the same system with the holes and gaps covered. Understanding the main noise paths and their contribution to the total noise across the frequency domain analyzed is critical for improving the acoustic performance of home appliances. This knowledge allows acousticians to apply different noise control techniques to meet standards or differentiate products from their competitors.

The results of this study could facilitate home appliance manufacturers in characterizing the acoustic performance of their products for different operating conditions by measuring only a few experimental setups. Radiation factor characterization only needs to be performed once for each washing machine, regardless of its operating configuration.

Although the radiation of the washing machine only contributes a small fraction of the total noise, it is crucial to study it along with the other noise paths. This is necessary because other configurations, such as using rigid gaskets or dry friction shock absorbers, may significantly increase the vibration levels of the cabin, and, therefore, impact overall noise.

Moreover, the presented methodology enables the analysis of each side of the washing machine separately, providing guidance on where noise control techniques would be more effective and where they would not significantly improve the acoustic performance. Overall, the findings of this study have significant implications for improving the acoustic performance of washing machines and can inform future research in this area.

**Author Contributions:** Conceptualization, C.A. and B.S.-T.; Methodology, C.A.; software, C.A.; validation, C.A. and B.S.-T.; formal analysis, C.A. and B.S.-T.; writing—original draft preparation, C.A. and B.S.-T.; writing—review and editing, C.A. and B.S.-T. All authors have read and agreed to the published version of the manuscript.

**Funding:** This research received no external funding.

**Data Availability Statement:** Not applicable.

**Conflicts of Interest:** The authors declare no conflict of interest.

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
