# Peer review of "Experimental Methodology to Characterize the Noise Paths in a Horizontal-Axis Washing Machine"

_acoustics, doi:10.3390/acoustics5020028_

Round 1

Reviewer 1 Report

Albero and Sanchez presented an experimental investigation of noise assessment of a typical washing machine. They have explained each part of the washing machine and their noise emission cause. the manuscript is very informative and suitable for acoustic communities. Few suggestions:

> Some references can be cited While describing the noise sources and mechanisms.

> Line no 174: correct the reference power value.

> Measurement method can be simplified further for readers.

> If possible, a figure can be added to show the expected noise path of the washer.

I couldn't find any grammatical mistakes. However, a few sentences can be rewritten for clarification.

Author Response

Thank you so much for your interesting comments and suggestions, we appreciated them.  We have considered all of them and the manuscript has been modified accordingly.

Kind Regards Cristian

Reviewer 2 Report

This manuscript presents a noise characterization practice on a washing machine. The content is within this journal's scope and could serve as a guide for noise control engineers in choosing proper noise control techniques for an actual home appliance product. However, it is this reviewer's opinion that the work lacks sufficient scientific novelty to meet the journal's standards. In this work, the contribution of each noise path is identified by conducting noise and vibration measurements on two configurations, which will be a natural choice made by a noise control engineer. Therefore, authors should demonstrate the significance of their work compared with others.

Author Response

Thank you so much for your interesting comments and suggestion, we appreciated them.  We have considered your observation, and we have included some paragraphs describing our contributions in comparison with other authors like you suggested. Please, find our comments in the document attached.

Kind Regards Cristian

Reviewer 3 Report

Very interesting topic, also for the use of intensity in acoustics in which there are few papers in which it is possible to find these applications.

Were the measurements performed on only one type of machine?

How does the noise change as the load in the drum changes?

What type of washing did you study?

The problem is vibrations and noise transmission on the floor.

Berardi et al. (2017) studied the effects of vibrations of water-stop of washing machines, they used accelerometers.

Can you check how to reduce the noise emitted?

What happens at low frequencies, low frequencies are transmitted in homes, so it would be good to see what happens at 63 Hz as well.

Thanks for your attention

Author Response

Thank you so much for your interesting comments and questions, we appreciate them. We have tried to answer your questions in the document attached, please do not hesitate to contact us if you want more information about any of them.

Kind Regards

Cristian

Round 2

Reviewer 1 Report

The revised manuscript amended the queries and suggestions. Overall, the content of this work is up-to-date and may find interest among the readers.

Author Response

Thank you so much for your comments.

Kind Regards Cristian

Reviewer 2 Report

The authors explained the contributions and novelty of their work

Author Response

(The authors gave the same response as above.)
